# Evaluation of Selective COX-2 Inhibition and In Silico Study of Kuwanon Derivatives Isolated from *Morus alba*

**DOI:** 10.3390/ijms22073659

**Published:** 2021-04-01

**Authors:** Seung-Hwa Baek, Sungbo Hwang, Tamina Park, Yoon-Ju Kwon, Myounglae Cho, Daeui Park

**Affiliations:** 1Department of Predictive Toxicology, Korea Institute of Toxicology, Daejeon 34114, Korea; seunghwa.beak@kitox.re.kr (S.-H.B.); sungbo.hwang@kitox.re.kr (S.H.); tamina.park@kitox.re.kr (T.P.); 2Center for Convergent Research of Emerging Virus Infection, Korea Research Institute of Chemical Technology, Daejeon 34114, Korea; 3Department of Human and Environmental Toxicology, University of Science and Technology, Daejeon 34113, Korea; 4National Institute for Korean Medicine Development, Gyeongsan 38540, Korea; mars005@nikom.or.kr (Y.-J.K.); meanglae@nikom.or.kr (M.C.)

**Keywords:** kuwanon A, cyclooxygenase inhibition assay, docking simulation, quantum mechanics

## Abstract

Six kuwanon derivatives (A/B/C/E/H/J) extracted from the roots of *Morus alba* L. were evaluated to determine their cyclooxygenase (COX)-1 and 2 inhibitory effects. Cyclooxygenase (COX) is known as the target enzyme of nonsteroidal anti-inflammatory drugs (NSAIDs), which are the most widely used therapeutic agents for pain and inflammation. Among six kuwanon derivatives, kuwanon A showed selective COX-2 inhibitory activity, almost equivalent to that of celecoxib, a known COX inhibitor. Kuwanon A showed high COX-2 inhibitory activity (IC_50_ = 14 μM) and a selectivity index (SI) range of >7.1, comparable to celecoxib (SI > 6.3). To understand the mechanisms underlying this effect, we performed docking simulations, fragment molecular orbital (FMO) calculations, and pair interaction energy decomposition analysis (PIEDA) at the quantum-mechanical level. As a result, kuwanon A had the strongest interaction with Arg120 and Tyr355 at the gate of the COX active site (−7.044 kcal/mol) and with Val89 in the membrane-binding domain (−6.599 kcal/mol). In addition, kuwanon A closely bound to Val89, His90, and Ser119, which are residues at the entrance and exit routes of the COX active site (4.329 Å). FMO calculations and PIEDA well supported the COX-2 selective inhibitory action of kuwanon A. It showed that the simulation and modeling results and experimental evidence were consistent.

## 1. Introduction

Nonsteroidal anti-inflammatory drugs (NSAIDs) are the most widely used therapeutic agents to manage pain and inflammation. NSAIDs exert their pharmacological action by inhibiting cyclooxygenase (COX) enzymes, which catalyze the conversion of arachidonic acid to prostaglandins, prostacyclins, and thromboxanes [1]. COX-1, an enzyme in a constitutive form, is expressed in several tissues and is known to play an important role in the synthesis of cytoprotective prostaglandins in the gastrointestinal tract. COX-2, a predominantly induced form of the enzyme, is also constitutively expressed in many tissues such as the prostate, endothelium, brain, and renal medulla [2,3]. COX-3, a protein derived from COX-1, is mainly found in the heart and cerebral cortex [4]. COX-1 is a housekeeping enzyme that is widely expressed in most tissues, whereas COX-2, mainly expressed at sites of infection, inflammation, and cancer, produces prostanoids which are responsible for disease pathogenesis. The two COX isoforms, COX-1 and COX-2, are NSAID targets [5]. The therapeutic anti-inflammatory action of NSAIDs is associated with the inhibition of COX-2; however, undesired side effects arise from the inhibition of COX-1 [6]. Therefore, there is a need for research on anti-inflammatory substances having an improved COX-1/2 selectivity index.

*Morus alba* L. belongs to a genus of flowering plants of the family Moraceae. It is widespread in Japan, India, China, and Korea, as well as in Europe, North and South America [7]. *Morus alba* L. contains many biologically active compounds such as steroids [8], terpenoids [9], alkaloids [10], stilbenes [11], coumarins [12], and flavones such as kuwanon derivatives [13,14,15]. Because of their high flavonoid content, the leaves, fruits, and bark of *Morus alba* L. have long been used to prepare medicines, especially for the treatment of cutaneous inflammation [16], psychotics [17], sputum [18], and asthma [19]. Although the recent study reported the inhibitory activity of constituents of *Morus alba* L. as COX inhibitors, the COX-1/2 selectivity index on kuwanon derivatives has not yet been studied [20].

COX-2 protein consists of an N-terminal epidermal growth factor-like domain, a membrane-binding domain, a peroxidase active site, and a COX active site [21]. The membrane-binding domain contains four α-helices that form a hydrophobic surface, which bind to a single leaflet of the lipid bilayer on the luminal side of the endoplasmic reticulum and the nuclear envelope [22]. The membrane-binding domain also forms the entrance of a narrow hydrophobic channel, which is the COX active site [23]. The membrane-binding domain and the COX active site are separated by three conserved residues, Arg120, Tyr355, and Glu524, which act as the gate of the COX active site [21,22,23,24]. This active site forms a small and narrow pocket that extends approximately 25 Å into a globular catalytic domain that is about 8 Å wide on average [23], and to which only small molecules like celecoxib can bind.

In the present study, six kuwanon derivatives were isolated from *Morus alba* L., and their selective inhibitory activities against COX-1/2 were investigated. To study the molecular interaction between COX-2 and kuwanon derivatives, ab initio quantum mechanical (QM) calculations were performed, including the fragment molecular orbital (FMO) method [25] and pair interaction energy decomposition analysis (PIEDA) [26], which provide accurate molecular interaction information based on wave function. For the FMO method, we applied the density-functional tight-binding (DFTB) method which is an efficient semi-empirical QM method expected to provide reasonable accuracy [27].

## 2. Results

### 2.1. Isolation and Identification of Compounds

Six natural products were isolated from Mori cortex (roots of *Morus alba* L.) and identified by comparing HPLC peak profiles (Appendix A) and NMR data (Appendix A) with literature values: kuwanon A [28,29], kuwanon B [28,29], kuwanon C [28,29,30], kuwanon E [31], kuwanon G [32], and kuwanon H [32]. The structures of the six compounds are shown in Figure 1. 

### 2.2. Analysis of COX-1/2 Enzyme Inhibitory Activity

To evaluate the activity of the kuwanon derivatives compared to that of celecoxib, the inhibitory activity of the isolated compounds against ovine COX-1 and COX-2 was evaluated (Table 1). Kuwanon A and G did not inhibit COX-1 action up to 100 μM. Kuwanon A showed a reasonable COX-2 inhibitory activity in vitro, with an IC_50_ of 14 μM and a selectivity index of >7.1; it was the most potent inhibitor among the kuwanon derivatives, with effects nearly similar to those of celecoxib.

### 2.3. Docking Simulation

To determine the binding pose of the kuwanon derivatives on COX-2, we performed docking simulation except for kuwanon G, because it had no COX-2 inhibition activity. Since the molecular weight and volume for the kuwanon derivatives were higher than those of celecoxib (Table 2), the binding site of the kuwanon derivatives was used as the binding pocket of the podophyllotoxin structure in chemocoxib A. The pocket of podophyllotoxin represents the gate of the COX active site consisting of Arg120 and Tyr355. The top 20 docking scores for the kuwanon derivatives and the root mean squared distance (RMSD) from the best binding pose are listed in Appendix A. All binding poses for kuwanon derivatives are described in Appendix A. To determine various conformation for binding poses, we superimposed to all binding poses for each kuwanon derivative and calculated average RMSD between the best binding poses and other poses (Appendix A). The best binding poses of kuwanon derivatives were located close to the gate of the COX active site; the configurations which were located more than 10 Å from each best binding pose were removed.

### 2.4. Interaction Energy in the Gate of COX Active Site

To analyze the molecular interactions between COX-2 and the kuwanon derivatives, we performed FMO calculations and PIEDA based on their docking complexes. The total interaction energy between COX-2 and top 20 binding poses for kuwanon derivatives are listed in Appendix A. The binding poses for the kuwanon derivatives are described in Figure 2. All kuwanon derivatives were bound to close to the gate of the COX active site consisting of Arg120 and Tyr355. In addition, the interaction energy between kuwanon derivatives and Arg120, and Tyr355 were also calculated by FMO and PIEDA (Figure 3). Interestingly, the pair interaction energies between two amino acids and kuwanon derivatives were closely related to their COX-2 inhibition activity. The Pearson-correlation coefficient between IC_50_ and the summation of pair interaction energies was calculated to be 0.920. As a result, the COX inhibition activity of kuwanon derivatives was related to their interactions with Arg120 and Tyr355.

### 2.5. Interaction Energy in the Membrane-Binding Domain and the Entrance/Exit Routes of COX Active Sites

To understand the differences in COX-2 inhibition activity among kuwanon derivatives, we calculated the pair interaction energies between amino acids consisting of the membrane-binding domain and kuwanon derivatives using FMO and PIEDA. The positive control compound used the podophyllotoxin of chemocoxib A bound to the membrane-binding domain of COX-2 (PDB ID: 4OTJ). Among amino acids consisting of the membrane-binding domain, five amino acids were selected as the lowest pair interaction energies (Table 3). The amino acids were Lys79, Lys83, Val89, Leu117, and Tyr122. In the binding site, the His122 in human COX-2 was Tyr122 in murine COX-2. (Appendix A). The structure of chemocoxib A bound to COX-2 is described in Figure 4. 

Next, we compared the pair interaction energy between residues in the entrance and exit routes of the COX active site and kuwanon derivatives. In the previous study [21], Val89, His90, and Ser119 played a major role as the entrance/exit routes of the COX active site. Interestingly, the pair interaction energies between Val89 and kuwanon A (−6.599 kcal/mol) and H (−6.257 kcal/mol) were higher than the other derivatives (Table 3). 

We also calculated the average closest distance from Val89, His90, and Ser119, located on the entrance/exit route, to the kuwanon derivatives, to determine how well these derivatives blocked this route. The closest distances between these residues and the kuwanon derivatives are listed in Table 4. 

The average closest distance from these residues is as follows: 4.329 Å for kuwanon A, 4.812 Å for kuwanon B, 4.303 Å for kuwanon C, 4.430 Å for kuwanon E, and 3.592 Å for kuwanon H, the closest to these residues. Although kuwanon C is located closer to Val89 residue than kuwanon A, the pair interaction energy score with Val89 was a positive energy score (EX score: 8.521 kcal/mol, Table 3) for kuwanon C. This means that kuwanon C could have steric repulsion interaction with Val89. The pair interaction energy of Val89 supports that kuwanon A and kuwanon H have higher COX-2 inhibition activity than other derivatives.

## 3. Discussion

Six kuwanon compounds isolated from the roots of *M. alba* L. were tested with an in vitro COX-1/2 inhibition assay. Kuwanon A and H demonstrated better COX-2 inhibitory activity than celecoxib. Kuwanon A exhibited better inhibitory activity against COX-2 compared to COX-1, with an excellent COX-2 selectivity index of >7.1. These results showed that kuwanon A was a potent inhibitor of COX-2 enzymes, even better than celecoxib. In addition, kuwanon H was observed to inhibit COX-1 although kuwanon H has higher COX-2 inhibition activity than other derivatives. As gastrointestinal disease, bleeding, and increased cardiovascular risk are known side effects of COX-1 inhibition [33], the development of COX-2 selective inhibitors is important.

The molecular weights and volumes of the kuwanon derivatives are larger than celecoxib but smaller than chemocoxib A. Since the size of the COX active site is similar to that of celecoxib, kuwanon derivatives could not bind to the COX active site like celecoxib. The residues Arg120 and Tyr355 at the gate of the COX active site are important residues for binding by NSAIDs. When there is no ligand in the COX active site, the gate of the COX active site is in its closed form; however, the gate of the COX active site converts to the open form when arachidonic acid, the COX substrate, approaches the binding site. In a previous study, the conformation change for the gate of the COX active site was determined using molecular dynamics simulation [34]. This conformational change increases the size of the gate of the COX active site and becomes a structure capable of ligand binding. This conformational change could also be confirmed by superimposing murine and human COX (Appendix A). In addition, Arg120, Tyr355, and Glu524 interact with the ligand through hydrogen bonding, stabilizing it in the active site [23,24,34]. For these reasons, the interaction between Arg120/Tyr355 and the kuwanon derivatives plays an important role in determining their COX-2 inhibitory activity. As a result, both kuwanon A and H showed strong interactions with Arg120 and Tyr355 in the gate of the COX active site (interaction energies of −7.044 kcal/mol and −8.863 kcal/mol, respectively). The Pearson correlation coefficient between IC_50_ and the summation of pair interaction energies was calculated to be 0.920.

A previous study suggested that Val89, His90, and Ser119 play a major role as the entrance and exit route of the COX active site [21]. Also, Val89 is a member of the membrane binding domain of COX protein. All kuwanon derivatives interacted strongly with Lys83 and Val89, with an absolute pair interaction energy greater than 4 kcal/mol (Table 3). Interestingly, the pair interaction energies between Val89 and kuwanon A (−6.599 kcal/mol) and H (−6.257 kcal/mol) were higher than other derivatives. In addition, the interaction between kuwanon derivatives and Lys83 was primarily through electrostatic and hydrophobic interactions. Electrostatic interactions with Lys83 are formed through the oxygen atom in the alcohol or ketone group of the kuwanon derivatives. Although kuwanon C showed high energy scores in electrostatic and hydrophobic interactions, it was also calculated to have a strong steric repulsion with Val89 since the distance between a hydrogen atom in a benzene-1,3-diol group and Val89 was calculated to be small. Therefore, kuwanon A and kuwanon H are more stably bound to COX-2 than other derivatives and these derivatives also have high COX-2 inhibition activity.

## 4. Materials and Methods

### 4.1. Materials

The roots of *Morus alba* L. were purchased from a commercial herbal market on May 2014 at Yeongcheon, Gyeongbuk, Korea. The organic solvents such as methanol (MeOH), chloroform (CHCl_3_), ethyl acetate (EtOAc), acetone, and n-hexane (Hx) were purchased from Duksan Chemical Co. (Seoul, Korea). Silica gel 60 (Merck 70–230 mesh, 230–400 mesh, ASTM, Germany) and octadecyl silica gel (ODS-A, 12 nm, S-150 m, YMC, Tokyo, Japan) were used for column chromatography. The NMR spectra were recorded on a JEOL ECX-500 spectrometer, operating at 500 MHz for 1 h and 125 MHz for the ^13^C NMR spectrum (JEOL Ltd., Tokyo, Japan). The high performance liquid chromatography (HPLC) system was an Agilent 1260 series (Agilent Inc., Santa Clara, CA, USA) equipped with a quaternary pump, a degasser, an injector, a column thermostat, a diode array detector (DAD), and an evaporative light scattering detector (ELSD).

### 4.2. Extraction and Isolation

The roots of *M. alba* L. (6 kg) were extracted three times with MeOH for 24 h at room temperature (3 × 10 L) to obtain a crude MeOH extract. The crude MeOH extract (299 g) was suspended in 2 L distilled water, and solvent partitioning was done with the same volume of Hx, EtOAc, and H_2_O. The EtOAc soluble fraction (110 g) was subjected to silica gel (Kieselgel 60, 70–230 mesh, Merck, Germany) column chromatography with a gradient of CHCl3/MeOH (20:1 to 0:1) to produce seven fractions (MAE 1–7). The MAE 4 fraction was separated into 13 fractions (MAE 4–1 to 4–13) by silica gel column chromatography with CHCl3/acetone (50:1 to 1:1). The subfractions MAE 4–4 and 4–5 were further purified on reversed phase (ODS-A) column chromatography with MeOH/H_2_O (1:9 to 9:1) to obtain **kuwanon A (1)** (102 mg, purity: 95.2%), **kuwanon B (2)** (164 mg, purity: 99.9%), **kuwanon C (3)** (26 mg, purity: 99.9%), and **kuwanon E (4)** (62 mg, purity: 99.9%). The fraction MAE 5 was isolated by silica gel column chromatography with CHCl_3_/acetone (30:1 to 1:1) to produce 8 subfractions (MAE 5–1 to 5–8). The subfractions MAE 5–3 and 5–4 were again subjected to chromatography using a silica gel column with CHCl_3_/MeOH (50:1 to 1:1), and then further purified on a reversed phase (ODS-A) column chromatography using MeOH/H2O (1:9 to 9:1) to obtain **kuwanon G (5)** (900 mg, purity: 98.0%), and **kuwanon H (6)** (193 mg, purity: 98.9%) (Figure 1). The structures of the six compounds obtained were determined by comparing their spectroscopic data with those in the literature (1–6). The determination of HPLC chromatograms was performed on an Agilent 1260 series system (Agilent Inc., Santa Clara, CA, USA) equipped with a binary pump, an autosampler, a column oven, a phenomenex kinetex C_18_ column (2.6 μm, 150 × 4.6 mm, Phenomenex), a photodiode array detector, and an evaporative light scattering detector (ELSD). The solvent used was 0.1% (*v/v*) trifluoroacetic acid (TFA) in water (solvent A) and 0.1% (*v*/*v*) TFA in acetonitrile (solvent B). The gradient elution conditions used were from 95% A/5% B at 3 min to 0% A/100% B at 30 min. The flow rate was 0.5 mL/min, and the injection volume was 3 μL.

### 4.3. Assay Materials 

Dimethyl sulfoxide (DMSO) and COX-1/2 Inhibitor Screening Kit (BioVison, California, CA, USA) were used according to the manufacturer’s instructions. All other reagents used were of the highest purity commercially available. All samples were dissolved in DMSO at a concentration of 100 mM. The chemical structures were drawn using ChemOffice software (http://www.cambridgesoft.com (accessed on 15 August 2020)).

### 4.4. In Vitro COX-1/COX-2 Inhibition Assay

The ability of the test compounds to inhibit COX-1 and COX-2 isoenzymes in vitro was determined using a Biovision fluorometric COX-1/-2 inhibitor screening assay kit. Briefly, the COX assay buffer (75 μL), COX cofactor working solution (2 μL), COX probe solution (1 μL), recombinant COX-1 or COX-2 (1 μL), arachidonic acid/NaOH solution (10 μL), and test solution (10 μL) were mixed in a 96-well plate, and the fluorescence kinetics were measured for 10 min at 25 °C. The fluorescence of each well was measured with an excitation wavelength of 535 nm and an emission wavelength of 587 nm. SC560 and celecoxib were used as positive controls for the COX-1 and the COX-2 assays, respectively. Fluorescence was measured using a microplate spectrophotometer (Bio-Tek, Winooski, VT, USA). Two points (T_1_ and T_2_) in the linear range of the plot were used to obtain the corresponding fluorescence values (RFU_1_ and RFU_2_). Then, the slope for all samples (S) was calculated by dividing the net ΔRFU (RFU_2_ − RFU_1_) values by the time ΔT (T_2_ − T_1_) using the following equation:
Relative inhibition (%) = ((slope of the enzyme control − slope of S)/(slope of the enzyme control)) × 100.

The concentration of the test compound causing 50% inhibition (IC_50_, μM) was calculated from the concentration inhibition response curve. The selectivity indices (SI, COX-1 IC_50_/COX-2 IC_50_) were also calculated and compared with that of the standard COX-2 selective inhibitor, celecoxib. The samples were evaluated in triplicates at five different concentrations.

### 4.5. Docking Simulation

The target protein, human COX-2 (PDB ID: 5IKR), was obtained from the Protein Data Bank (PDB) database (https://www.rcsb.org/ (accessed on 15 September 2020)). The 3D structures of the kuwanon derivatives, kuwanon A (PubChem CID: 44258296), kuwanon B (PubChem CID: 44258295), kuwanon C (PubChem CID: 5481958), kuwanon E (PubChem CID: 10342292), kuwanon G (PubChem CID: 5281667), and kuwanon H (PubChem CID: 5281668), were prepared for docking simulation. Since kuwanon G and kuwanon H only had 2D structures, we used the ChemOffice program to convert them into 3D structures with the minimum energy. The conformational evaluation of the 3D structure generated with the energy minimization method using ChemOffice is described in the Appendix A. The pocket of COX-2, known as the COX active site, was not the binding site of celecoxib, but the binding site of chemocoxib A [24], because the molecular weight and volume for the kuwanon derivatives were bigger than those for celecoxib. The definition of the binding site was described in the Appendix A in detail. Chemocoxib A is a podophyllotoxin-indomethacin conjugate. Indomethacin, which is smaller than podophyllotoxin, is located in the COX active site. However, podophyllotoxin is bound to the membrane-binding domain. Therefore, the binding pocket of the kuwanon derivatives was defined using the superimposed binding sites from the podophyllotoxin-bound crystal structures of mouse COX-2 (PDB ID: 4OTJ). To consider conformational variability, we performed molecular docking simulation using AutoDock Vina [35]. AutoDock Vina has been applied to ligand-flexible docking. We have confirmed that AutoDock Vina provides sufficient conformational variability, which is described in the Appendix A. The binding poses were displayed in Chimera [36]. The molecular weights and volumes of celecoxib, chemocoxib A (podophyllotoxin-indomethacin conjugate), and kuwanon derivatives were calculated using RDKit [37].

### 4.6. FMO Calculation and PIEDA

The two-body FMO method was applied to all calculations in this work for the FMO2/DFTB method. All input files were prepared in compliance with the hybrid orbital projection scheme fragmentation. The two cysteine residues forming the disulfide bond were defined as one fragment. The other parameters calculated in the FMO calculation were default values while the total charges for the kuwanon conformers were defined as zero. The FMO calculation and PIEDA were performed with the 30 June 2020 R1 GAMESS version (https://www.msg.chem.iastate.edu/gamess/download.html (accessed on 15 August 2020)) [38]. 

PIEDA was performed based on the second-order Møller–Plesset perturbation theory and polarizable continuum model level with the 3–21G basis set. PIEDA can explain the molecular interaction between two fragments using the decomposition method. These molecular interactions include electrostatic (ES), exchange repulsion (EX), charge transfer (CT), dispersion (DI), and solvation (SL). ES and EX are represented by salt bridges, hydrogen bonds, and polar interactions; DI is represented as hydrophobic interactions, and EX is represented as steric repulsion. The pair interaction energy was calculated to find significant COX-2 residues interacting with the kuwanon derivatives as well as podophyllotoxin, which was bound in the membrane-binding domain. The significant residues were found with the Pearson-correlation coefficient between experimental inhibitory activity (IC_50_) and pair interaction energy.

## 5. Conclusions

Among the six kuwanon compounds isolated from the roots of *Morus alba* L., kuwanon A showed the highest COX-2 selective inhibitory activity, even higher than celecoxib. Kuwanon derivatives cannot bind to the COX active site because kuwanon derivatives are larger than celecoxib. However, kuwanon derivatives could inhibit by effectively blocking the gate of the COX active site including Arg120 and Tyr355. 

Interestingly, kuwanon A had the strongest interaction with Arg120 and Tyr355 at the gate of the COX active site (−7.044 kcal/mol) and with Val89 in the membrane-binding domain (−6.599 kcal/mol). In addition, kuwanon A closely bound to Val89, His90, and Ser119, which are residues at the entrance and exit routes of the COX active site (4.329 Å). As both kuwanon A and kuwanon H interact strongly with Val89, they have higher COX-2 inhibitory activity than the other derivatives.

## Figures and Tables

**Figure 1 ijms-22-03659-f001:**
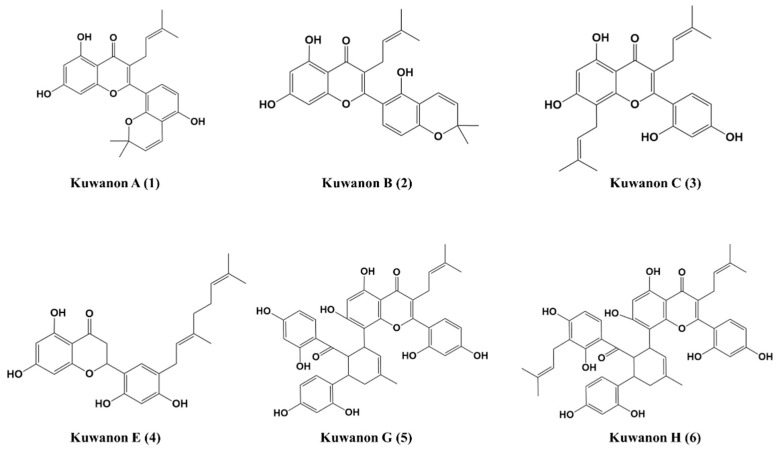
Chemical structures of the kuwanon derivatives isolated from the roots of *Morus alba* L.

**Figure 2 ijms-22-03659-f002:**
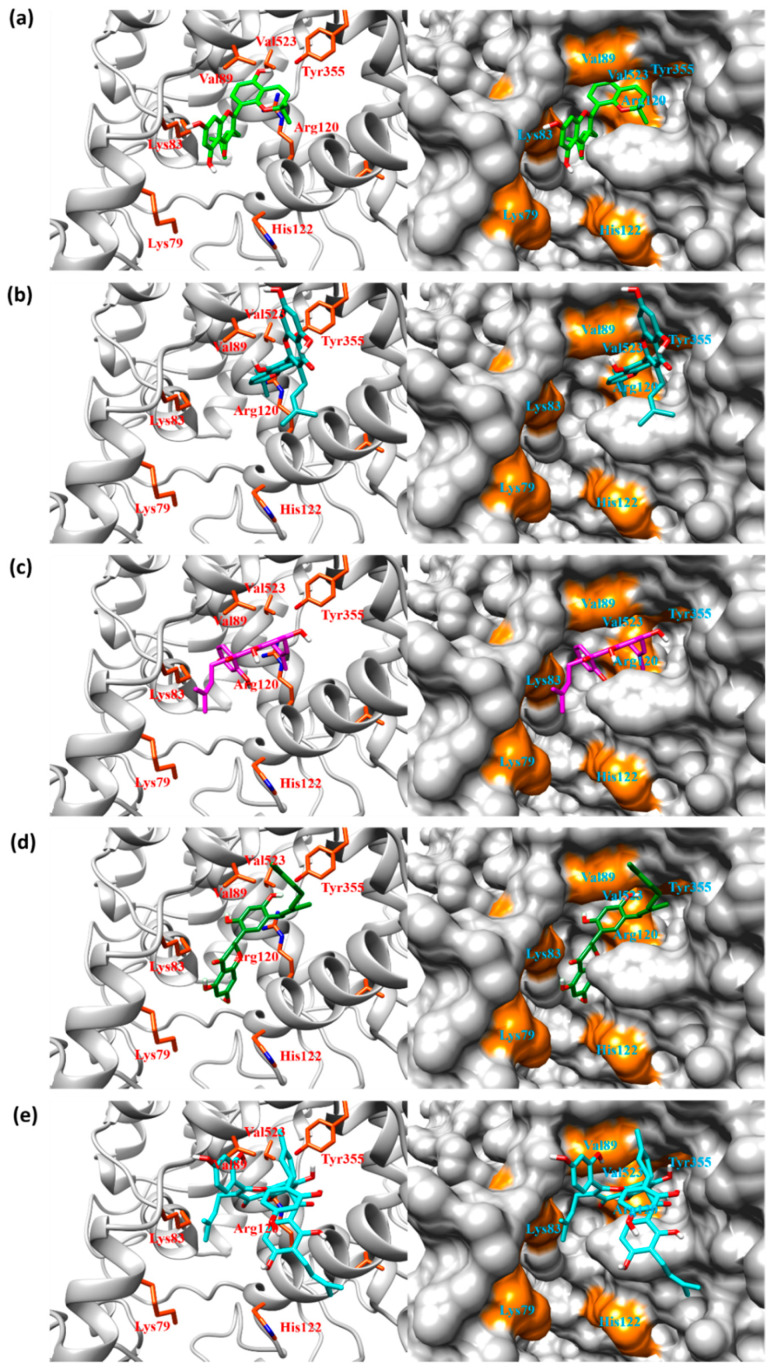
Binding poses with the lowest total interaction energy for (**a**) kuwanon A, (**b**) kuwanon B, (**c**) kuwanon C, (**d**) kuwanon E, and (**e**) kuwanon H. All kuwanon derivatives can block the entrance of the celecoxib active site. The orange-colored surface represents the residues at the gate of the cyclooxygenase (COX) activity as calculated with the five lowest pair interaction energies, using pair interaction energy decomposition analysis (PIEDA).

**Figure 3 ijms-22-03659-f003:**
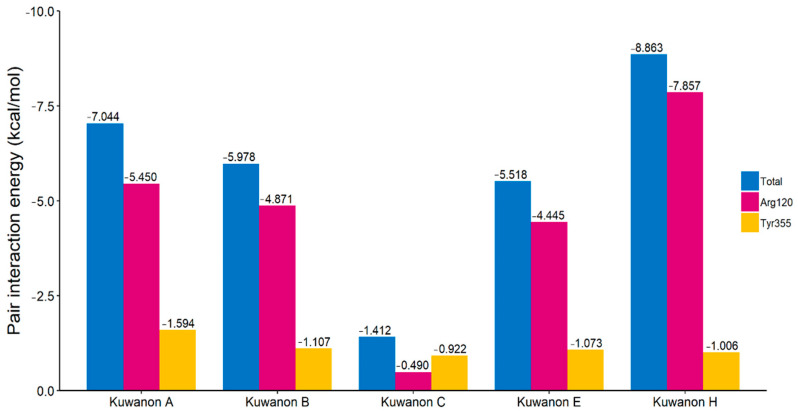
The pair interaction energies between the kuwanon derivatives and the two amino acids in the gate of the COX active site. Both Arg120 and Tyr355 play a major role at the gate of the COX active site.

**Figure 4 ijms-22-03659-f004:**
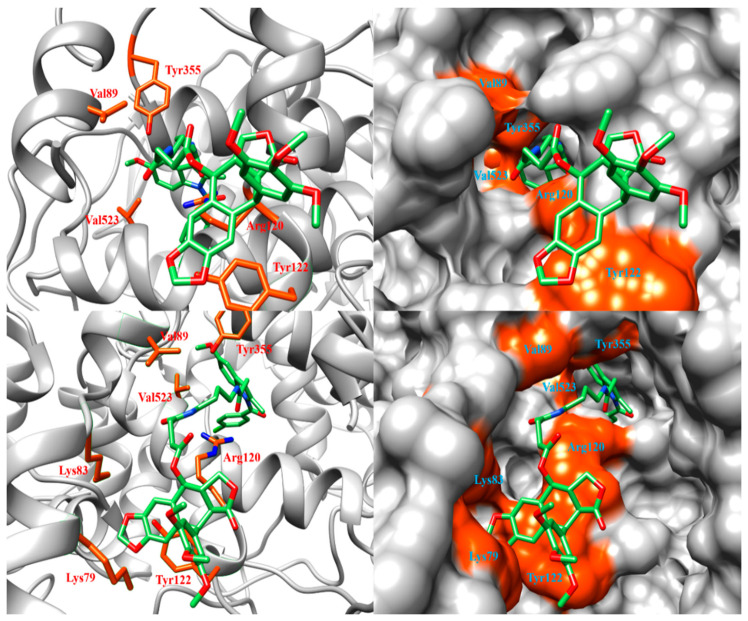
Binding pose between chemocoxib A and COX-2 (PDB ID: 4OTJ). The orange-colored surface represents the residues at the gate of COX activity. Since the binding pose was hidden by COX-2 residues, the upper figure is represented by removing residues from Thr71 to Lys83, and the lower figure was represented by removing residues from Tyr115 to Ser119.

**Table 1 ijms-22-03659-t001:** Results of the in vitro COX-1/2 enzyme inhibition assay.

Compounds	IC_50_ (μM) ^1^		SI ^2^
	COX-1	COX-2	
Kuwanon A	>100	14	>7.1
Kuwanon B	36	28	1.3
Kuwanon C	67	42	1.6
Kuwanon E	46	34	1.4
Kuwanon G	>100	>100	- ^3^
Kuwanon H	37	7	5.0
Celecoxib	>100	16	>6.3

^1^ IC_50_ is the concentration required to produce 50% inhibition of the COX-1 or COX-2 enzyme assay. ^2^ SI (selectivity index, COX-1 IC_50_/COX-2 IC_50_). ^3^ Not determined.

**Table 2 ijms-22-03659-t002:** Molecular weight and volume for COX-2-bound drug, components of chemocoxib A (indomethacin and podophyllotoxin), and kuwanon derivatives.

Name	Molecular Weight (g/mol)	Molecular Volume
Celecoxib	381.076	359.848
Chemcoxib A	923.303	891.808
Indomethacin	357.077	350.384
Podophyllotoxin	456.142	434.096
Kuwanon A	420.157	417.768
Kuwanon B	420.157	413.624
Kuwanon C	422.173	426.152
Kuwanon E	424.189	427.352
Kuwanon G	692.226	662.832
Kuwanon H	760.288	744.432

**Table 3 ijms-22-03659-t003:** PIEDA of the five amino acids in the membrane-binding domain of COX-2.

Compound Name	Amino Acid	Pair Interaction Energy (kcal/mol)	Component of Pair Interaction Energy (kcal/mol) ^c^				
			ES	EX	CT	DI	SL
Chemocoxib A ^a^	Lys79	−15.267	−1.364	−0.126	−0.195	−5.647	−7.936
	Lys83	−10.241	−2.342	0.303	−0.257	−3.839	−4.105
	Tyr122	−6.049	0.502	0.104	0.000	−6.321	−0.334
	Leu117	−3.464	−3.101	0.001	0.000	−0.565	0.201
	**Val89**	**−3.267**	0.018	−0.072	−0.002	−3.200	−0.012
Kuwanon A ^b^	Lys79	−1.807	−2.344	0.000	0.000	−0.531	1.068
	Lys83	−14.430	−7.995	0.782	−0.152	−8.657	1.591
	His122	−0.830	−0.825	0.000	0.000	−0.258	0.253
	Leu117	−0.934	−1.120	0.756	−0.022	−0.609	0.061
	**Val89**	**−6.599**	−0.771	0.532	−0.016	−6.172	−0.172
Kuwanon B ^b^	Lys79	−1.468	0.839	0.000	0.000	−0.037	−2.271
	Lys83	−1.721	4.021	0.000	0.000	−0.503	−5.239
	His122	−0.096	0.069	0.000	0.000	−0.051	−0.114
	Leu117	−0.024	−0.103	0.000	0.000	−0.370	0.449
	**Val89**	**−4.665**	1.035	0.217	−0.011	−4.866	−1.040
Kuwanon C ^b^	Lys79	−3.164	0.158	−0.003	0.000	−1.035	−2.285
	Lys83	−14.541	−7.330	3.374	−0.198	−7.612	−2.775
	His122	−0.830	−0.825	0.000	0.000	−0.258	0.253
	Leu117	−0.934	−1.120	0.756	−0.022	−0.609	0.061
	**Val89**	**4.212**	−0.154	8.521	−0.128	−3.934	−0.091
Kuwanon E ^b^	Lys79	−2.662	−5.799	0.000	0.000	−0.382	3.518
	Lys83	−8.604	−3.086	0.050	−0.001	−5.367	−0.200
	His122	−1.933	−1.678	−0.001	0.000	−0.775	0.521
	Leu117	−0.199	−0.139	0.000	0.000	−0.351	0.291
	**Val89**	**−5.810**	−0.328	0.563	−0.023	−5.754	−0.267
Kuwanon H ^b^	Lys79	−3.997	−0.092	0.000	0.000	−0.434	−3.471
	Lys83	−10.573	−6.387	1.361	−0.033	−4.870	−0.645
	His122	−0.515	−0.150	0.000	0.000	−0.088	−0.277
	Leu117	1.646	2.200	0.000	0.000	−0.349	−0.205
	**Val89**	**−6.257**	0.117	0.334	−0.002	−6.521	−0.185

^a^ Pair interaction energy of chemocoxib A represents the interaction energy between podophyllotoxin and linker structures and murine COX-2 (PDB ID: 4OTJ). ^b^ Pair interaction energy of kuwanon derivatives represent the interaction energy between the kuwanon derivatives and human COX-2 (5IKR). ^c^ ES: ElectroStatic interaction, EX: EXchanged repulsion, CT: Charge Transfer, DI: Dispersion, SL: SoLvation. The bold amino acid is located in entrance/exit routes of COX-2 active sites.

**Table 4 ijms-22-03659-t004:** The closest distance between the kuwanon derivatives and amino acids in the entrance/exit routes of COX active sites.

Residues	Closest Distance (Å)				
	Kuwanon A	Kuwanon B	Kuwanon C	Kuwanon E	Kuwanon H
Val89	2.595	2.195	1.404	1.961	2.142
His90	3.951	5.109	5.205	3.664	4.857
Ser119	6.440	7.132	6.301	7.665	3.778
Average	4.329	4.812	4.303	4.430	3.592

## Data Availability

Data are contained within the article.

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
