# Peer review of "Evaluation of Selective COX-2 Inhibition and In Silico Study of Kuwanon Derivatives Isolated from Morus alba"

_ijms, 2021, doi:10.3390/ijms22073659_

Round 1

Reviewer 1 Report

The manuscript by Baek et al. "Evaluation of selective COX-2 inhibition and in silico study of kuwanon derivatives isolated from Morus alba" shows interesting data, experimental and modeling, on a series of compounds and rationalization of or their activities.

The work is carried out well and can be published as is

Author Response

Open Review_1

Comments and Suggestions for Authors

The manuscript by Baek et al. "Evaluation of selective COX-2 inhibition and in silico study of kuwanon derivatives isolated from Morus alba" shows interesting data, experimental and modeling, on a series of compounds and rationalization of or their activities.

The work is carried out well and can be published as is

  • Thank you for your review. We tried to edit English langue with repetitive and confusing sentences in the manuscript.

Reviewer 2 Report

The manuscript reports a study of the inhibitory effects of six natural products (kuwanon derivatives) targeting the cyclooxygenase enzymes COX-1 and COX-2. The empirically measured inhibition activity was explained in terms of molecular interactions derived from the in silico calculation - docking simulations, fragment molecular orbital (FMO) calculations, and pair interaction energy decomposition analysis (PIEDA) at the quantum-mechanical level. The use of these calculation techniques is explained by their reasonable calculation cost.

I’m agree with the idea that the computational cost can be a metric for the choice of methods, but it is not the main factor, especially in the health-related domains, there the accuracy of the results is absolutely required. In addition, each calculation must be carefully explained at the level of theory.

The Authors not only used the simplified methods for estimating the binding energy, but also these methods were applied to targets and inhibitors represented as wrong 3D models.

Indeed, the simple replacement (noted on line 154 as convertion) of the residue in murine COX-2 is not sufficient to obtain a correct human enzyme as the target required for docking. Similarly, the 3D models of inhibitors were obtained by the ChemOffice which converts 2D diagrams into 3D structures. This primitive modelling followed by energy minimization (see Methods) is not appropriate approach to this type of molecules which are characterised by a high conformational variability due to their large number of rotational conformers and the great flexibility of the aliphatic chains.

If the 3D models of a target and its inhibitors are  constructed incorrectly, further application of computational methods, simplified of advanced, is not reasonable and loses all meaning.

Reviewer 3 Report

The manuscript, concerning several prenylated flavonoids from Morus alba L. as COX-1 and -2 inhibitors, is quite interesting since highlights, for the first time, that, among the studied derivatives, Kuwanon A is endowed with a promising  COX-2 inhibiting activity (showing a very good selectivity index), that is supported by docking simulations, FMO calculations, and PIEDA at the quantum-mechanical level. 

The referee suggest the following minor corrections :

-to check the initial of compound names (for instance lines 19, 29, 85, 159  etc. Kuwanon A; lines 22, 69, 92, 97  etc. Celecoxib, lines 150, 154, 307 Chemocoxib A, etc.) into the whole text, including the Figure 2 legend

-to avoid words repetitions (lines 42/44, 77/78, 84/85, 142/143)

-to substitute in vitro with in vitro (lines 95, 99, 275)

-to uniform Table 4 caption character

-to distance OH and CO groups in Kuwanon H structure (Figure 1)

-to substitute isolates with isolated compounds (line 93)

-to uniform the references: to eliminate Volume (line 370) and LP (line 419), to check authors name (lines 395, 400) or paper title (lines 414 and 415) character

-to complete references (for instance: ref. 14, 17, 33 etc.)

-to add the missing authors (ref. 38)

-to uniform the text reporting Supplementary Materials (lines 84/85 and first page of Supplementary Materials).
